# RNAGenScape: Property-guided Optimization and Interpolation of mRNA Sequences with Manifold Langevin Dynamics

## Abstract

mRNA design and optimization are important in synthetic biology and therapeutic development, but remain understudied in machine learning. Systematic optimization of mRNAs is hindered by the scarce and imbalanced data as well as complex sequence-function relationships. We present `RNAGenScape`, a property-guided manifold Langevin dynamics framework that iteratively updates mRNA sequences within a learned latent manifold. `RNAGenScape` combines an *organized autoencoder*, which structures the latent space by target properties for efficient and biologically plausible exploration, with a *manifold projector* that contracts each step of update back to the manifold. `RNAGenScape` supports property-guided optimization and smooth interpolation between sequences, while remaining robust under scarce and undersampled data, and ensuring that intermediate products are close to the viable mRNA manifold. Across three real mRNA datasets, `RNAGenScape` improves the target properties with high success rates and efficiency, outperforming various generative or optimization methods developed for proteins or non-biological data. By providing continuous, data-aligned trajectories that reveal how edits influence function, `RNAGenScape` establishes a scalable paradigm for controllable mRNA design and latent space exploration in mRNA sequence modeling.

## 1 Introduction

Messenger ribonucleic acids (mRNAs) design and optimization are important (Qin et al., 2022; Metkar et al., 2024) but remain understudied in machine learning (Castillo-Hair & Seelig, 2021; Schlusser et al., 2024). Even small edits to mRNA sequences can strongly affect their stability, translation efficiency, and eventual protein output (Zhang et al., 2023; Li et al., 2025). For instance, modifying the non-coding 5' untranslated region (UTR) can tune the degradation rate of transcripts and therefore regulate protein production (Castillo-Hair et al., 2024; Ma et al., 2024). Such guided controls are directly relevant to applications such as mRNA vaccines (Pardi et al., 2018; Chaudhary et al., 2021) and protein replacement therapies (Qin et al., 2022; Vavilis et al., 2023), where improving translation efficiency or stability can improve efficacy and reduce dosage. However, systematic optimization in the mRNA space remains an open challenge, because (1) viable mRNAs occupy only a narrow subset of the vast ambient design space (Zhang et al., 2023; Calvanese et al., 2024), (2) data collected in this field are scarce and imbalanced, with many regions sparse or undersampled (Taubert et al., 2023; Asim et al., 2025), and (3) the sequence-function relationships are highly complex (Licatalosi & Darnell, 2010; Weinreb et al., 2016).

We present `RNAGenScape`, a property-guided Manifold Langevin dynamics framework dedicated for mRNA sequence design and optimization. `RNAGenScape` consists of two core modules: **1** an organized autoencoder (OAE) that learns the manifold of mRNA sequences and organizes the space by the target property, enabling efficient exploration within a biologically plausible subspace rather than the ambient sequence space; and **2** a manifold projector that contracts each update back onto the learned manifold, preserving biological plausibility. By preprocessing the data with SUGAR (Lindenbaum et al., 2018)

to augment undersampled regions by filling "holes" in the manifold, we ensure that the manifold projector remains effective even under sparse training conditions (e.g., as few as 2,000 data points in one dataset), addressing a common challenge in mRNA data.

By operating directly on the latent manifold rather than in the ambient sequence space, `RNAGenScape` is able to optimize target properties of the mRNAs, smoothly interpolate between sequences, and ensure that all intermediate results remain close to the viable mRNA manifold. Across three real-world mRNA datasets that span two orders of magnitude in size, `RNAGenScape` consistently improved target properties with high success rates and efficiency, outperforming existing approaches originally developed for proteins or generic sequence data.

In summary, our main contributions are as follows.

1. **Framework**: We propose `RNAGenScape`, a manifold Langevin dynamics framework that enables interpolation and continuous property-guided optimization of mRNA sequences starting from real data points, offering biologically grounded sequence modeling.

2. **Manifold constraint**: We introduce a learned manifold projector that ensures biological plausibility throughout optimization trajectories.

3. **Efficiency**: Unlike diffusion-based models which typically start from Gaussian noise and explore the entire Euclidean space, we restrict our exploration to the manifold and start from existing sequences, allowing faster training and inference.

4. **Empirical validation:** We provide results on three real mRNA datasets, demonstrating that our method improves target properties while maintaining manifold fidelity, outperforming various optimization and generation methods.

## 2 Preliminaries

### 2.1 Manifold hypothesis and manifold learning

The **manifold hypothesis** (Cayton et al., 2008; Narayanan & Mitter, 2010; Fefferman et al., 2016) posits that high-dimensional data lie near a low-dimensional manifold embedded in the ambient space. Formally, each observation $x_i \in \mathbb{R}^n$ arises from a smooth nonlinear map $\mathbf{f} : \mathcal{M}^d \to \mathbb{R}^n$ applied to a latent variable $z_i \in \mathcal{M}^d$, where $d \ll n$.

**Manifold learning** methods seek to recover this latent structure by constructing representations that preserve intrinsic geometry (Van Dijk et al., 2018; Moon et al., 2019; Burkhardt et al., 2021; Liu et al., 2024; Liao et al., 2024; Liu et al., 2025a;b; Sun et al., 2025). Diffusion geometry (Coifman & Lafon, 2006; Van Dijk et al., 2018; Lindenbaum et al., 2018) provides one such paradigm, where local similarities are defined via an anisotropic kernel on the pairwise similarities, and a Markov transition probability matrix is obtained by row normalization. This diffusion process encodes the intrinsic geometry of the data.

A point is considered *on-manifold* if it lies within the range of the nonlinear map $\mathbf{f}$, while *off-manifold* points deviate from this structure and may correspond to invalid or adversarial samples (Rifai et al., 2011; Li et al., 2023). Thus, projecting updated points back to the manifold is critical for robustness and geometry-aware optimization (He et al., 2023b).

**Stochastic gradient descent (SGD) on Riemannian manifolds** (Bonnabel, 2013) extends classical SGD by computing updates in the tangent space and mapping them back to the manifold via exponential maps or retractions. However, such methods assume an analytic form of the manifold. In contrast, the manifold underlying biological sequence space is not known in a closed form. `RNAGenScape` addresses this by directly *learning* the projection operator, enabling optimization on data manifolds without requiring analytic solutions.

### 2.2 Langevin-dynamics and beyond

**Diffusion Models** (Ho et al., 2020) are generative frameworks that learn a data distribution $p(x)$ by reversing a fixed Markov diffusion process of length $T$. Starting from Gaussian noise, they are trained to iteratively denoise samples through a sequence of learned denoising functions over $T$ steps. The training objective $\mathcal{L}_{DM} := \mathbb{E}_{x,\epsilon \sim \mathcal{N}(0,1),t} \left[ ||\epsilon - \epsilon_\theta(x_t, t)||_2^2 \right]$ is a reweighted form of the variational lower bound, closely related to denoising score matching (Song et al., 2021).

**Latent Diffusion Models** (Rombach et al., 2022) present an extension of the concept. Instead of performing the reverse diffusion process in the data space, they operate in a latent space after embedding the data with an encoder $\mathcal{E}$, where $z = \mathcal{E}(x)$. The modified objective is given by $\mathcal{L}_{\text{LDM}} := \mathbb{E}_{z,\epsilon \sim \mathcal{N}(0,1),t}\left[||\epsilon - \epsilon_\theta(z_t, t)||_2^2\right]$.

**Langevin Dynamics** (Song & Ermon, 2019) has been employed in generative models to sample from high-dimensional data distributions using only an estimate of the score function $\nabla_x \log p(x)$. In particular, it first trains a neural network $s_\theta$ to approximate the score function of data injected with Gaussian noise. Sampling is then performed via annealed Langevin dynamics, given by $\tilde{x}_t = \tilde{x}_{t-1} + \frac{\eta_i}{2}s_\theta(\tilde{x}_{t-1}, \sigma_i) + \sqrt{\eta_i}z_t$. Here, $s_\theta(\tilde{x}_{t-1}, \sigma_i)$ is the learned score function at noise level $\sigma_i$, and $\eta_i$ is the step size at that level. By gradually annealing from high to low noise, this procedure enables generation of high-quality samples without an explicit likelihood or energy model.

**Neural Stochastic Differential Equations (neural SDEs)** (Kidger et al., 2021), are differential equations simultaneously modeling two terms: a drift term $f(\cdot)$ depicting the true time-varying dynamics of the variable, and a diffusion term $g(\cdot)$ representing stochasticity using the Brownian motion $W_t$. The update rule is given by $\mathrm{d}X_t = f(t, X_t)\mathrm{d}t + g(t, X_t) \circ \mathrm{d}W_t$. From a high level, Langevin dynamics is a special case of neural SDEs after discretization.

## 3    RNAGenScape

The key components of our framework are ❶ **OAE**: an autoencoder module whose latent space is organized by the target property (Section 3.1, Figure 1a), ❷ **Manifold Projector**: a module that brings the updated latent embeddings back to the learned data manifold during each step of optimization/interpolation (Section 3.2, Figure 1b & 1d), and ❸ **Property-guided manifold Langevin dynamics**, a procedure that integrates the two aforementioned modules to enable property optimization and interpolation (Section 3.3 & 3.4, Figure 1e).

Once trained, these components allows `RNAGenScape` to optimize the target property of a given sequence (Section 4.3) and interpolate between existing sequences (Section 4.5).

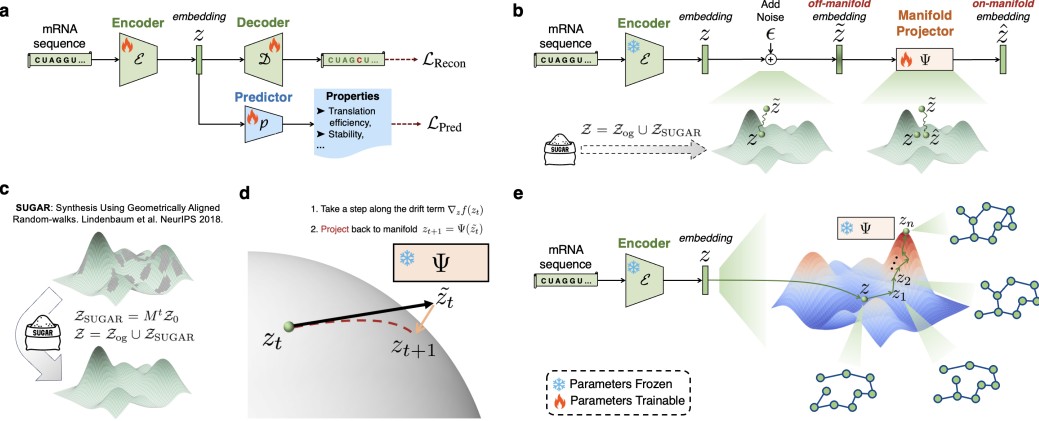

Figure 1: Schematic of `RNAGenScape`. **(a)** We first train an organized latent space for mRNA sequences by jointly optimizing reconstruction and property prediction objectives. **(b)** We then train a manifold projector while the encoder's weights are frozen. **(c)** For undersampled mRNA manifolds, we use SUGAR to learn key dimensions in the manifold and fill undersampled regions. **(d)** During optimization, the manifold projector brings off-manifold points back to the manifold. **(e)** We can use the encoder and the manifold projector to optimize the properties of given input mRNA sequences or interpolate between sequences. Notably, the intermediate products can also be decoded. Best viewed zoomed in.

### 3.1 Learning a latent space organized by property

We begin by training an **organized autoencoder (OAE)**, where the latent space is implicitly structured via supervision from a property prediction task (Figure 1a). Similar to a vanilla autoencoder (Hinton & Salakhutdinov, 2006), the encoder $\mathcal{E}$ maps the input mRNA sequence $x$ to a latent representation $z$, which is decoded by $\mathcal{D}$ back to the sequence space. In addition to this standard architecture, a predictor $\mathcal{P}$ infers properties $\hat{y}$ from the embedding $z$. Formally, $z_i = \mathcal{E}(x_i)$, $\hat{x}_i = \mathcal{D}(z_i)$, $\hat{y}_i = \mathcal{P}(z_i)$.

The latent space $\mathcal{Z}$ is thus shaped by jointly optimizing the reconstruction loss and the prediction loss (equation 1), encouraging it to learn sequence-relevant information while being organized by the target properties. $\lambda_{\text{Pred}}$ and $\lambda_{\text{Recon}}$ are hyperparameters that balance between organizing the property landscape and capturing sequence information. They are empirically set to 1 and 5 in all experiments. Here, $x_i \in \mathbb{R}^V$ is the ground truth one-hot encoding of the nucleotide at position $i$ in sequence $x$ with an mRNA vocabulary of size $V$.

$$\mathcal{L}_{\text{OAE}} = \lambda_{\text{Pred}}\mathcal{L}_{\text{Pred}} + \lambda_{\text{Recon}}\mathcal{L}_{\text{Recon}} = \lambda_{\text{Pred}} \overbrace{\mathbb{E}_{(x,y)\sim p_{\text{data}}}||\hat{y}_i - y_i||_2^2}^{\text{optimize } \mathcal{P} \text{ with MSE}} -\lambda_{\text{Recon}} \overbrace{\mathbb{E}_{x\sim p_{\text{data}}} \log \frac{\exp(\hat{x}_{i,x_i})}{\sum_{v=1}^{V} \exp(\hat{x}_{i,v})}}^{\text{optimize } \mathcal{E} \text{ and } \mathcal{D} \text{ with CrossEntropy}} \quad (1)$$

### 3.2 Training a manifold projector

**Learning the data manifold**   mRNA datasets are typically scarce, undersampled, and biased toward specific experimental conditions, which makes it difficult to learn a faithful latent manifold directly from the train data. To address this, we adopt SUGAR (Lindenbaum et al., 2018), a diffusion geometry-based generative method that learns the geometry of the data and samples the manifold uniformly. This augmentation enriches the latent space with geometry-preserving samples, helping the model better approximate the underlying mRNA manifold even in sparse regions.

Specifically, this preprocessing step yields an expanded latent set: $\mathcal{Z} = \mathcal{Z}_{\text{og}} \cup \mathcal{Z}_{\text{SUGAR}}$, $\mathcal{Z}_{\text{SUGAR}} = M^t \mathcal{Z}_0$, where $\mathcal{Z}_{\text{og}}$ are the original latent embeddings, $\mathcal{Z}_0$ are locally-sampled neighbors, and $M^t$ is the sparsity-corrected Markov diffusion transition matrix applied for $t$ steps.

**Learning the Manifold Projection**   To keep the generated trajectories aligned with the latent data manifold, we introduce a manifold projector $\Psi$. As illustrated in Figure 1b and magnified in Figure 1d, $\Psi$ takes in a noisy optimized point $\tilde{z}$ and projects it back onto or near the manifold.

To train $\Psi$, we adopt a denoising objective that contracts noisy samples back towards the clean points on the latent manifold. Given a clean latent embedding $z$, we construct a short corruption chain $\tilde{z}^{(0)} = z$, $\tilde{z}^{(k)} \sim C(\tilde{\mathcal{Z}}|\tilde{\mathcal{Z}}^{(k-1)}, \sigma_k)$, $k = 1, \ldots, K$, where $C(\cdot, \sigma_k)$ denotes Gaussian corruption with noise level $\sigma_k$. The projector is trained to reverse each step by predicting $\tilde{z}^{(k-1)}$ from $\tilde{z}^{(k)}$, yielding the objective in equation 2.

---

**Algorithm 1** Manifold Projector

**Input:** Dataset $\mathcal{Z} = \{z_i\}_{i=1}^N$, denoiser $\Psi$, noise levels $\{\sigma_1, \ldots, \sigma_K\}$, denoising steps $K$, learning rate $\eta$
**for** each $z_i$ in minibatch $\{z_i\}_{i=1}^B \subset \mathcal{Z}$ **do**
    Initialize $\tilde{z}^{(0)} \leftarrow z_i$
    **for** $k = 1$ to $K$ **do**
        $\tilde{z}^{(k)} \sim C(\tilde{\mathcal{Z}}|\tilde{\mathcal{Z}}^{(k-1)}, \sigma_k)$
        $\mathcal{L}^{(k)} = \|\Psi(\tilde{z}^{(k)}) - \tilde{z}^{(k-1)}\|_2^2$
    **end for**
    $\mathcal{L}_i = \sum_{k=1}^K \mathcal{L}^{(k)}$
    $\Psi \leftarrow \Psi - \eta \nabla_\Psi \left( \frac{1}{B} \sum_{i=1}^B \mathcal{L}_i \right)$
**end for**

---

$$\mathcal{L}_\Psi = \mathbb{E}_{z\sim p_{\text{data}}} \sum_{k=1}^{K} \mathbb{E}_{\tilde{z}^{(k)}\sim C(\cdot|\tilde{z}^{(k-1)}, \sigma_k)} \left[ \|\Psi(\tilde{z}^{(k)}) - \tilde{z}^{(k-1)}\|_2^2 \right] \quad (2)$$

When $K = 1$, this reduces to the standard denoising autoencoder loss (Vincent et al., 2008). In practice, we keep $K$ small (e.g. 1-3) to capture *local updates near the data manifold*, rather than simulating long diffusion chains from Gaussian noise. As a result, our algorithm is fast during training and inference.

### 3.3 Property-guided manifold Langevin dynamics

Next, we introduce a novel property-guided manifold Langevin-dynamics framework.

Given a trained encoder $\mathcal{E}$, a property predictor $\mathcal{P}$, and a manifold projector $\Psi$, our Langevin-dynamics framework optimizes sequences for a target property. Starting from the latent embedding $z = \mathcal{E}(x)$ of a sequence $x$, we iteratively update it using a gradient-based drift term $\nabla_z f(z)$, inject Gaussian noise $\epsilon$, and apply a manifold projection $\Psi(\cdot)$ to ensure biological plausibility and interpretability. We define the update rule as described in equation 3.

$$z_{t+1} = \Psi(z_t + \mathrm{d}z_t), \quad \mathrm{d}z_t = \frac{\eta}{\tau}\nabla_z f(z_t) + \sqrt{2\eta}\,\epsilon_t, \quad \epsilon_t \sim \mathcal{N}(0, I) \tag{3}$$

Here, $\eta$ is the step size, $\tau$ is the temperature hyperparameter, and $\nabla_z f(z)$ denotes the property gradient given by the predictor $\mathcal{P}$. A smaller $\tau$ emphasizes focused updates along the gradient, while a larger $\tau$ encourages more diverse exploration. When $\tau \to \infty$, the property guidance vanishes and the update rule is dominated by the stochastic term, which becomes similar to generative modeling with the walkback algorithm (Bengio et al., 2013).

The manifold projector $\Psi$, analogous to the retraction in Riemannian SGD (Bonnabel, 2013), is applied after each update to ensure that each step remains near the biologically valid latent manifold, enabling interpretable and controllable generation trajectories.

With the trained components $\mathcal{E}$, $\mathcal{P}$ and $\Psi$, we can optimize the target property of any given sequence. Notably, optimization entails both maximization and minimization: users can choose to increase or decrease the target property, depending on the application.

### 3.4 Interpolating between sequences

Beyond property optimization, our framework also enables *interpolation* between existing mRNA sequences by guiding the latent embedding of one sequence toward that of another. Specifically, given a source sequence $x_{\text{source}}$ and a target sequence $x_{\text{target}}$, we first obtain their latent embeddings via the encoder: $z_{\text{source}} = \mathcal{E}(x_{\text{source}})$ and $z_{\text{target}} = \mathcal{E}(x_{\text{target}})$.

Starting from $z = z_{\text{source}}$, we run property-guided manifold Langevin dynamics with an additional force term that pulls the latent toward $z_{\text{target}}$. The interpolation force is defined as $f_{\text{interp}}(z, z_{\text{target}}) = -\frac{z - z_{\text{target}}}{\|z - z_{\text{target}}\|_2}$, which provides a normalized directional bias toward the target point. Incorporating this into the Langevin update changes the drift term while all other components remain intact, as described in equation 4.

$$\mathrm{d}z_t = \frac{\eta}{\tau}f_{\text{interp}}(z_t, z_{\text{target}}) + \sqrt{2\eta}\,\epsilon_t \tag{4}$$

This modification steers the latent trajectory smoothly toward the target, enabling interpretable interpolations between biological sequences.

## 4 Empirical Results

In this section, we demonstrate the effectiveness of `RNAGenScape` on two key tasks: (1) mRNA sequence optimization and (2) mRNA sequence interpolation. The first task is broadly relevant to applications in therapeutics and synthetic biology. For example, enhancing the translation efficiency and stability of an mRNA vaccine can increase its protein yield and persistence, thereby boosting therapeutic efficacy while reducing the required dose. The second task facilitates the exploration of intermediate variants. This can provide insights into the functional landscape of regulatory elements within mRNAs of interest.

### 4.1 Experimental Settings

**Datasets and tasks** We evaluate `RNAGenScape` on three mRNA datasets that capture diverse contexts and experimental designs. The optimization objectives are underlined.

1. `Zebrafish` includes five subsets of zebrafish 5' UTR, experimentally measured using nascent protein-transducing ribosome affinity purification (Strayer et al., 2023), a massively parallel reporter assay to quantify translation control. It spans various stages and conditions of development, and each subset contains approximately 11,000 5' UTR sequences each with 124 nucleotides along with annotations on translation efficiency.

2. `OpenVaccine` contains 2,400 mRNA sequences devised for COVID-19 mRNA vaccines, each with 107 nucleotides (Das et al., 2020). They are collected with degradation profiles under multiple conditions, quantified to mRNA stability relevant to vaccine design.
3. `Ribosome-loading` is a large-scale library of approximately 260,000 5' UTR sequences each with 50 nucleotides (Sample et al., 2019), paired with pseudouridine-modified coding sequences of enhanced green fluorescent protein. The sequences are annotated on mean ribosome load, a property that reflects translation efficiency.

**Baselines**   We compared our method with a range of popular *de novo* generative modeling approaches, including variational autoencoder (VAE) (Kingma et al., 2013), Wasserstein generative adversarial network with gradient penalty regularization (WGAN-GP) (Gulrajani et al., 2017), denoising diffusion probabilistic model (DDPM) (Ho et al., 2020), latent diffusion model (LDM) (Rombach et al., 2022), and flow matching (FM) (Rombach et al., 2022). We also included classic optimization methods, including gradient ascent (Williams, 1992; Zinkevich, 2003), Markov chain Monte Carlo (MCMC) (Brooks, 1998; Andrieu et al., 2003), and hill climbing (Selman & Gomes, 2006). All classic optimization baselines were GPU-compatible adaptions from the implementation in (Castro et al., 2022). Lastly, we benchmarked against optimization methods originally designed for proteins, DiffAb (Luo et al., 2022), IgLM (Shuai et al., 2023), and NOS (Gruver et al., 2023).

**Evaluation**   Since the optimization process could and should result in mRNA sequences not covered by the dataset, to quantify their properties, we trained a separate property prediction model $\mathcal{P}_{\text{oracle}}(x)$ to serve as a proxy of the ground truth. $\mathcal{P}_{\text{oracle}}(x)$ is used for evaluation only, and is *strictly invisible during inference to avoid circular dependency*.

**Reproducibility**   All experiments were performed under 5 random seeds and the average results are reported. Hyperparameters and hardware used are summarized in Appendix A.

### 4.2  RNAGenScape PRODUCES STRUCTURED, DATA-ALIGNED TRAJECTORIES

`RNAGenScape` operates within a learned latent space that reflects the manifold of real biological sequences. To illustrate this behavior, we visualize individual optimization runs in Figure 2. The trajectory exhibits monotonic increases in the target property, while remaining near regions populated by real sequences. See Appendix D for more examples. These trajectories are direct consequences of the manifold-constrained dynamics, which guided each step toward high-property regions while staying on the manifold.

Importantly, all intermediate steps during optimization can be decoded into mRNA sequences, allowing researchers to examine how sequences evolve step by step as specific properties are

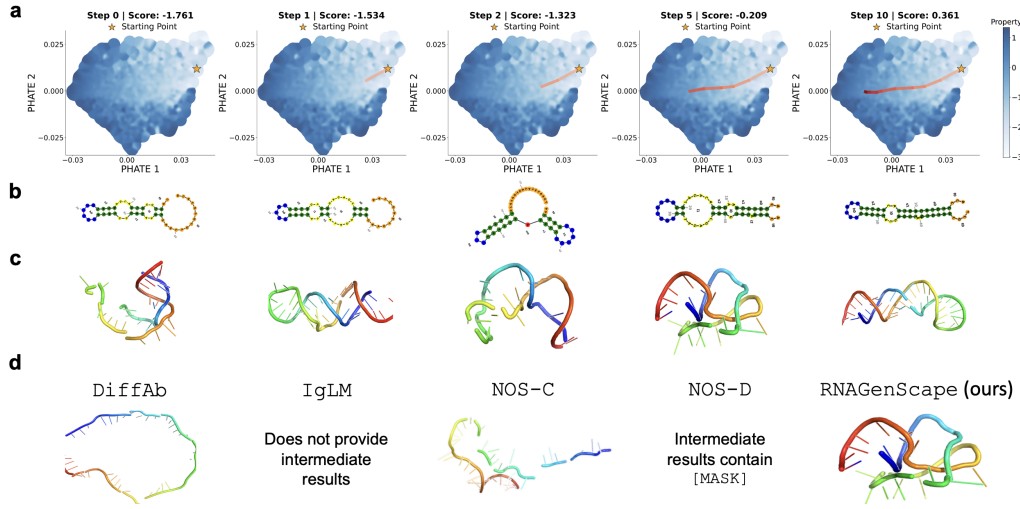

Figure 2: Latent space trajectories of `RNAGenScape` over 10 optimization steps. The trajectories follow smooth and reasonable paths with steady improvement in the target property. **(a)** Trajectories in the PHATE space. **(b)** 2D structures. **(c)** 3D structures. **(d)** Intermediate products of various methods midway through optimization (5th step).

Table 1: Our proposed `RNAGenScape` achieves superior property optimization while also being inference-efficient. Top performers among property optimization methods are bolded. For *de novo* generative models, the optimization columns are grayed out, as they cannot explicitly steer properties; reported values instead reflect their learned distributions. $\Delta$ denotes the median change in property. % denotes success rate (the percentage of mRNAs improved).

| Methods | Inference Speed | Zebrafish (n≈55k) property = translation efficiency | | | | OpenVaccine (n≈2k) property = stability | | | | Ribosome-loading (n≈260k) property = mean ribosome load | | | |
| | | +property | | −property | | +property | | −property | | +property | | −property | |
| | ms/sample ↓ | Δ ↑ | % ↑ | Δ ↓ | % ↑ | Δ ↑ | % ↑ | Δ ↓ | % ↑ | Δ ↑ | % ↑ | Δ ↓ | % ↑ |
|---|---|---|---|---|---|---|---|---|---|---|---|---|---|
| *de novo* generative models | | | | | | | | | | | | | |
| VAE (Kingma et al., 2013) | 0.13 | 0.80 | 67.7 | 0.80 | 32.3 | -0.41 | 40.0 | -0.41 | 60.0 | -0.37 | 38.3 | -0.37 | 61.7 |
| WGAN-GP (Gulrajani et al., 2017) | 0.07 | -1.23 | 16.4 | -1.23 | 83.6 | -0.02 | 47.9 | -0.02 | 53.1 | -1.16 | 22.5 | -1.16 | 77.5 |
| DDPM (Ho et al., 2020) | 0.91 | 0.16 | 55.4 | 0.16 | 44.6 | 0.28 | 64.6 | 0.28 | 35.4 | -0.28 | 40.4 | -0.28 | 59.6 |
| LDM (Rombach et al., 2022) | 0.74 | -0.43 | 36.5 | -0.43 | 63.5 | 1.38 | 78.0 | 1.38 | 21.0 | -1.11 | 46.0 | -1.11 | 54.0 |
| FM (Lipman et al., 2022) | 5.82 | 0.17 | 55.6 | 0.17 | 44.4 | 0.20 | 62.9 | 0.20 | 37.1 | -0.25 | 41.9 | -0.25 | 58.1 |
| property optimization methods | | | | | | | | | | | | | |
| DiffAb (Luo et al., 2022) | 41.04 | 0.20 | 62.4 | 0.17 | 41.1 | 0.37 | 73.8 | 0.40 | 26.2 | -1.0 | 41.5 | -0.16 | 60.8 |
| IgLM (Shuai et al., 2023) | 157.57 | 0.07 | 52.9 | 0.01 | 49.3 | 0.07 | 54.8 | 0.06 | 64.7 | 0.42 | 69.6 | -1.28 | 80.6 |
| NOS-C (Gruver et al., 2023) | 0.99 | -0.03 | 48.6 | -0.66 | 70.4 | 0.96 | 90.0 | -0.05 | 52.1 | -0.21 | 42.6 | -0.26 | 59.0 |
| NOS-D (Gruver et al., 2023) | 0.96 | 0.22 | 57.1 | 0.20 | 42.7 | 0.46 | 71.8 | 0.25 | 36.2 | -0.25 | 41.3 | -0.26 | 59.3 |
| Sequence-space MCMC | 3.84 | -0.53 | 33.5 | -0.54 | 67.1 | -0.13 | 41.8 | -0.18 | 58.7 | -1.02 | 25.0 | -1.56 | 83.8 |
| OAE + Gradient Ascent | **0.50** | -0.51 | 33.9 | -0.44 | 63.8 | 0.31 | 66.3 | -0.19 | 61.3 | -0.47 | 34.7 | -1.60 | 84.0 |
| OAE + MCMC | 10.93 | -0.43 | 35.8 | -0.44 | 64.2 | 0.17 | 40.0 | -0.17 | 60.0 | -1.41 | 18.7 | -1.42 | 81.3 |
| OAE + Hill Climbing | 81.52 | -0.52 | 33.6 | -0.56 | 67.5 | 0.16 | 40.8 | -0.16 | 60.0 | -1.38 | 19.0 | -1.39 | 81.0 |
| OAE + Stochastic Hill Climbing | 99.66 | -0.51 | 33.5 | -0.56 | 67.9 | 0.11 | 43.1 | -0.15 | 60.0 | -1.40 | 20.0 | -1.41 | 80.4 |
| RNAGenScape without SUGAR (ours)[1] | 0.57 | 1.19 | 89.3 | -0.83 | 74.2 | 1.39 | 95.2 | -0.33 | 65.2 | 0.46 | 72.4 | -1.66 | 87.3 |
| RNAGenScape (ours) | 0.57 | **1.48** | **94.0** | **-1.32** | **85.6** | 1.33 | 93.5 | **-0.87** | **86.9** | **0.46** | **72.4** | **-1.66** | **87.3** |

optimized. As a qualitative illustration of biological plausibility, we show that intermediate results can be properly folded by ViennaRNA (Lorenz et al., 2011) and RhoFold (Shen et al., 2024) into 2D and 3D structures (Figure 2b-c). In contrast, folding intermediate products of several other methods lead to failure, as indicated by the broken structures (Figure 2d).

### 4.3 RNAGenScape achieves superior property optimization

We quantitatively compare `RNAGenScape` against a range of *de novo* generative models and optimization baselines (Table 1). Although *de novo* approaches are effective in modeling the data distribution, they offer limited to no control over the target properties. As a result, their performance in property optimization is unfavorable.

Among property optimization methods, `RNAGenScape` consistently delivers the strongest results, achieving the highest median property change and the highest success rate in both the positive and negative directions. In particular, its median improvement is nearly twice that of the next-best method in several cases, and its success rate exceeds all other approaches.

In addition to being successful in optimizing properties, `RNAGenScape` also achieves the best manifold fidelity, as shown in Table 2. See Appendix B for more details on the evaluations.

Table 2: Manifold fidelity, represented by the average $\ell_2$ distance between the generated or optimized sequences to the data manifold. Results are averaged over all datasets.

| Methods | latent space distance ↓ |
|---|---|
| VAE | 0.737 |
| WGAN-GP | 1.729 |
| DDPM | 0.254 |
| LDM | 0.584 |
| FM | 0.259 |
| DiffAb | 0.237 |
| IgLM | 0.740 |
| NOS-C | 0.539 |
| NOS-D | 0.260 |
| Sequence-space MCMC | 0.292 |
| OAE + Gradient Ascent | 0.460 |
| OAE + MCMC | 0.297 |
| OAE + Hill Climbing | 0.300 |
| OAE + Stochastic Hill Climbing | 0.288 |
| RNAGenScape without SUGAR (ours) | **0.233** |
| RNAGenScape (ours) | 0.235 |

### 4.4 Efficiency and scalability

In addition to its strong property control, `RNAGenScape` is also highly efficient at inference time. As reported in Table 1, it achieves an inference speed of 0.57 ms/sample, nearly matching the fastest method (gradient ascent at 0.50 ms/sample) and substantially faster than many other property optimization methods (such as hill climbing at 81.53 ms/sample, DiffAb at 41.04 ms/sample, and IgLM at 157.57 ms/sample). This efficiency makes `RNAGenScape` well suited for large-scale or iterative design workflows where fast feedback is essential.

---

[1]The optimal SUGAR upsampling ratio is 0 for `Ribosome-loading`, and hence we have identical performance with and without SUGAR in that dataset. See ablation studies (Section 4.6).

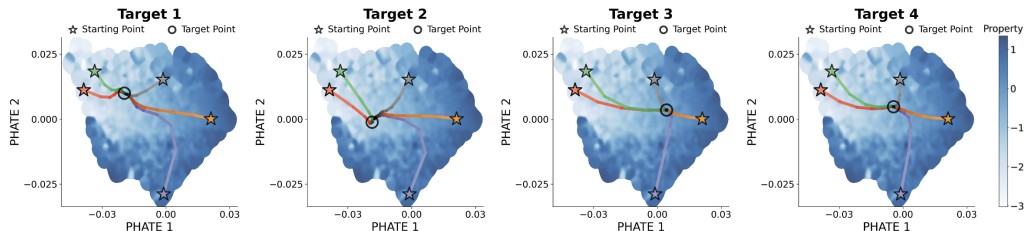

Figure 3: Latent space interpolation trajectories from 5 sources to 4 targets. Each trajectory is shown as a line fading from bright to dark in a consistent color. `RNAGenScape` produces smooth and coherent paths on the manifold between arbitrary input-target mRNA pairs.

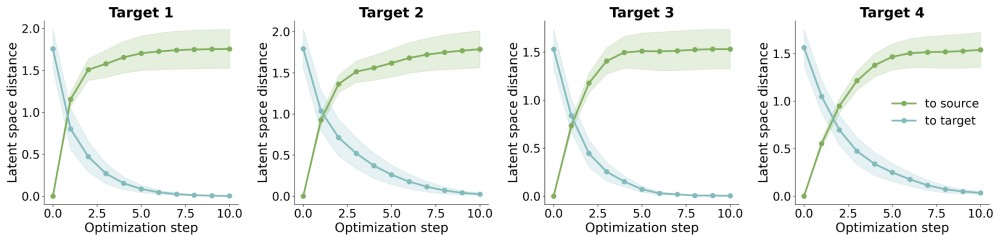

Figure 4: Latent space $\ell_2$ distances during interpolation show smooth and monotonic transition from the source to the target. Results are averaged over all data samples.

### 4.5 INTERPOLATING BETWEEN ARBITRARY SEQUENCES

`RNAGenScape` enables interpolation between arbitrary sequences using the directional drift term (equation 4). Guided by a directional force toward a specified target, `RNAGenScape` generates smooth and coherent trajectories on the learned manifold while preserving biological plausibility and continuity (Figure 3). These trajectories connect arbitrary input-target sequence pairs in a structured manner, reflecting semantically meaningful transitions.

The distances from each intermediate point to the source and target quantitatively demonstrate the monotonicity and smoothness of the interpolation (Figure 4).

### 4.6 ABLATION STUDIES

**Manifold projector** Our first ablation shows that the manifold projector $\Psi$, a core contribution of `RNAGenScape`, is essential for its performance (Table 3). Without $\Psi$, the method completely fails at property optimization. This outcome is

Table 3: Manifold projector $\Psi$ is critical.

| $\Psi$ | +property | | −property | | latent space distance ↓ |
| | $\Delta$ ↑ | % ↑ | $\Delta$ ↓ | % ↑ | |
| --- | --- | --- | --- | --- | --- |
| ✗ | -0.17 | 45.1 | 0.13 | 47.0 | 0.355 |
| ✓ | **0.46** | **72.4** | **-1.66** | **87.3** | **0.120** |

expected: following the property gradient without projecting back to the manifold causes trajectories to drift away, as reflected in the tripled latent space distance in Table 3.

**Optimization steps** Next, we analyze how sensitive `RNAGenScape` is to the number of optimization steps. The results in Figure 5 show that our method is able to converge quickly and remains stable over a range of Langevin dynamics steps.

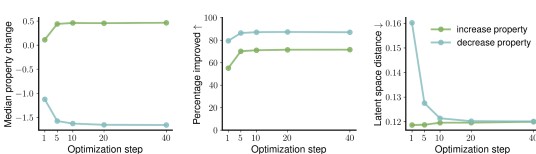

Figure 5: `RNAGenScape` optimization is step-efficient and remains stable over a range of optimization steps.

**SUGAR** To assess the necessity to enrich the latent space using SUGAR, we performed an ablation study to test whether and how much these geometry-preserving samples facilitate `RNAGenScape`. The experiments are summarized in Table S2. As expected, smaller datasets benefit from more aggressive manifold sampling (best ratio = 0.1 for `Zebrafish` (n≈55k), 1.0 for `OpenVaccine` (n≈2k), and 0.0 for `Ribosome-loading` (n≈260k)).

## 5    Conclusion

We introduced RNAGenScape, a property-guided manifold Langevin dynamics framework dedicated for mRNA design and optimization. By combining an organized autoencoder that aligns the latent space with target properties and a denoising-based manifold projector, RNAGenScape steers existing sequences along smooth, manifold-aligned trajectories that both improve target properties and preserve biological plausibility. Empirically, RNAGenScape outperforms various generative and optimization methods in property control and manifold fidelity, while matching or exceeding their inference efficiency. It also shows promises in faithfully interpolating between real biological sequences. With this work, we also hope to shift the paradigm of biological sequence design from unconstrained generation to guided optimization, and shine more light on mRNA sequence design as a critical yet understudied frontier in computational biology.

## 6    Limitations and Future Work

One limitation of our approach is its dependence on the fidelity of the organized latent space: if the organized autoencoder fails to capture critical sequence constraints, manifold projections may permit small but functionally invalid drifts. Additionally, our current formulation optimizes a single scalar property; extending RNAGenScape to multi-objective settings would broaden its applicability. Finally, while we have demonstrated compelling in silico gains, integrating real-world experimental feedback remains an important avenue to validate and refine the learned manifold.

In future work, We will study the possibility to perform sequence-structure joint modeling and optimization. Beyond mRNA, we plan to extend RNAGenScape to other modalities such as protein sequences and regulatory elements, and integrate active learning frameworks that guide wet lab experimentation. By grounding sequence optimization in the manifold of real data, we aim to provide a versatile platform for interpretable and high-throughput design in synthetic biology.

## 7    Related Works

Machine learning is becoming increasingly popular for optimizing biological sequences such as DNA, RNA, and proteins. This section reviews recent advances in sequence modeling and optimization, with an emphasis on mRNAs.

Current machine learning approaches for sequence design can be grouped into three main paradigms, but each addresses only part of the challenge (Table 4).

*De novo* generative models excel at creating novel sequences, but fundamentally operate by generating from scratch rather than refining existing functional

Table 4: Characteristics of the models: whether they can perform (1) generation, (2) optimization, (3) interpolation, and (4) whether they produce optimization trajectories.

| Method | Gen. | Opt. | Interp. | Traj. |
|---|---|---|---|---|
| *de novo* generative models | ✓ | ✗/✓ | ✗ | ✗ |
| classic optimization methods | ✗/✓ | ✓ | ✗ | ✓ |
| DiffAb (Luo et al., 2022) | ✓ | ✓ | ✗ | ✓ |
| IgLM (Shuai et al., 2023) | ✓ | ✓ | ✗ | ✗ |
| NOS (Gruver et al., 2023) | ✓ | ✓ | ✗ | ✗ |
| RNAGenScape (ours) | ✓ | ✓ | ✓ | ✓ |

sequences (Prykhodko et al., 2019; Méndez-Lucio et al., 2020; Dauparas et al., 2022; Wu et al., 2021; Madani et al., 2023; Watson et al., 2023). Classic optimization strategies (Williams, 1992; Zinkevich, 2003; Brooks, 1998; Andrieu et al., 2003; Selman & Gomes, 2006) are capable of improving known sequences, but typically lack mechanisms to ensure that intermediate variants remain consistent with the underlying biological distribution. More recent deep learning methods for sequence generation and optimization (Luo et al., 2022; Shuai et al., 2023; Gruver et al., 2023) aim to combine generative modeling with property-driven objectives, but their optimization trajectories remain opaque, offering limited interpretability of which sequence changes drive functional improvements. Furthermore, existing methods cannot interpolate between sequences. Consequently, a framework that enables biologically-grounded sequence engineering remains needed (Wu et al., 2019).

An extended related works section can be found in Appendix E.

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

# Technical Appendices for
# `RNAGenScape`: Property-guided Optimization and Interpolation of mRNA Sequences with Manifold Langevin Dynamics

## A  Hyperparameters and Architecture

**Learning the manifold with SUGAR**  To learn the manifold with SUGAR, we used the $k$-NN mode for estimating degrees (and thus sparsity) of latent points. We employed an $\alpha$-decay kernel with $\alpha = 2$ and an adaptive bandwidth determined from the distance to the 5 nearest neighbors. The diffusion time was set to $t = 1$.

**Training the organized autoencoder (OAE)**  We trained the OAE using the AdamW optimizer with an initial learning rate of $10^{-2}$, together with a linear warmup cosine annealing scheduler. The learning rate was linearly increased from $10^{-4}$ (i.e., $0.01\times$ the base learning rate) to the target value during the first 10% of training epochs (warmup), and then annealed to zero following a cosine decay schedule over the remaining epochs. We used a batch size of 128, a maximum of 200 epochs, and early stopping with a patience of 20 epochs based on the validation loss.

**Training the manifold projector**  We trained the manifold projector $\Psi$ using the AdamW optimizer with a learning rate of $10^{-4}$. We used a batch size of 256, a maximum of 200 epochs, and applied early stopping with a patience of 20 epochs based on the validation loss.

Table S1: Hyperparameters used for different datasets.

| Dataset | $\lambda_{\mathrm{Recon}}$ / $\lambda_{\mathrm{Pred}}$ | Noise levels | Langevin (step size, temperature) |
|---|---|---|---|
| Zebrafish | 5.0 / 1.0 | $\{1.0, 0.8, 0.5\}$ | $1 \times 10^{-2}$, $8 \times 10^{-3}$ |
| OpenVaccine | 1.0 / 1.0 | $\{1.0, 0.5\}$ | $1 \times 10^{-2}$, $1 \times 10^{-2}$ |
| Ribosome-loading | 1.0 / 1.0 | $\{0.3\}$ | $5 \times 10^{-3}$, $1 \times 10^{-2}$ |

**Organized Autoencoder (OAE)**  The organized autoencoder (OAE) maps mRNA sequences $x \in \mathbb{R}^{L \times V}$ to a compact latent $z \in \mathbb{R}^d$. Our latent dimension is 320 across all datasets.

For encoder, we apply three 1D convolutional blocks with GroupNorm, GELU, and channel squeeze-excitation (SE), followed by adaptive average pooling to length 8 and a linear projection. The property head is a three-layer MLP with GELU and dropout rate set to 0.3.

For decoding, a progressive 1D decoder upsamples structure gradually: we first expand $z$ to a $128 \times 8$ seed map, then apply a stack of UpsampleBlock modules composed of upsampling and two residual convolutional blocks until reaching $\geq L$ positions; we then refine the output with two residue convolutional blocks to produce the final predicted logits. Weights are Kaiming/Xavier initialized; GroupNorm scales are set to 1 and biases to 0.

**mRNA sequence vocabulary**  Although mRNA sequences naturally consist of the nucleotides `A`, `U`, `G`, and `C`, some experimental datasets represent `U` as `T` (borrowing the DNA alphabet). To handle this heterogeneity consistently, we define a unified vocabulary of size $V = 7$: `<pad>`, `A`, `U`, `T`, `G`, `C`, and `N`. Here, `N` denotes an unknown nucleotide during sequencing, and both `U` and `T` tokens are retained to ensure compatibility across datasets.

**Hardware**  The evaluations were performed on a single NVIDIA A100 GPU. However, `RNAGenScape` can be run efficiently on more modest hardware.

## B  EVALUATION METRICS

**Property Optimization**   To quantify the effectiveness of property optimization of different models, we measure both the median improvement and the fraction of sequences that are successfully optimized.

Specifically, given a test set of mRNA sequences $X_{\text{test}}$ with predicted properties $\mathcal{P}_{\text{oracle}}(X_{\text{test}})$ and their optimized counterparts $\tilde{X}_{\text{test}}$ with properties $\mathcal{P}_{\text{oracle}}(\tilde{X}_{\text{test}})$, we compute:

$$\Delta_{\text{median}} = \text{median}\left(\mathcal{P}_{\text{oracle}}(\tilde{x}) - \mathcal{P}_{\text{oracle}}(x)\right), \quad x \in X_{\text{test}}, \tag{5}$$

and

$$\%_{\text{success}} = \frac{1}{|X_{\text{test}}|} \sum_{x \in X_{\text{test}}} \mathbb{1}\left[\mathcal{P}_{\text{oracle}}(\tilde{x}) > \mathcal{P}_{\text{oracle}}(x)\right]. \tag{6}$$

Here, $\Delta_{\text{median}}$ measures the improvement in the target property across the test set, while $\%_{\text{success}}$ reports the percentage of sequences that improve after optimization.

For models that cannot refine existing sequences (e.g., pure *de novo* generators), we assign a random pairing between initial and final sequences to enable a fair comparison.

**Manifold Fidelity**   Given a property prediction model $\mathcal{P}_{\text{oracle}}$ with an encoder $\mathcal{E}_{\text{oracle}}$, a test set of mRNA sequences $X_{\text{test}}$, and generated or optimized sequences $\tilde{X}_{\text{test}}$, we quantify manifold fidelity as the average minimum $\ell_2$ distance in the latent space of $\mathcal{E}_{\text{oracle}}$ between each new sample and the test data:

$$\mathcal{M}_{\text{fidelity}} = \mathbb{E}_{\tilde{x} \sim \tilde{X}_{\text{test}}}\left[\min_{x \in X_{\text{test}}} \left\|\mathcal{E}_{\text{oracle}}(\tilde{x}) - \mathcal{E}_{\text{oracle}}(x)\right\|_2\right]. \tag{7}$$

This metric captures how closely the generated or optimized sequences remain to the empirical data manifold defined by $X_{\text{test}}$.

# C   ABLATION ON SUGAR

We performed ablation on the upsampling ration in SUGAR and the results are summarized in Table S2. In theory, SUGAR is most helpful in datasets that show greater sparsity and more undersampled regions on the data manifold.

In our ablation studies, we observe that different datasets require different ideal SUGAR ratios. In general, smaller datasets need higher upsampling to achieve the optimal performance. While dataset size not necessarily reflect the density or sparsity of the data manifold, in general they seem to be positively correlated. In future practice, we suggest using higher SUGAR upsampling ratio when working with smaller datasets, which is very common in the world of mRNA design.

Table S2: Ablation on SUGAR.

| SUGAR ratio | Zebrafish (n≈55k) | | | | OpenVaccine (n≈2k) | | | | Ribosome-loading (n≈260k) | | | |
|---|---|---|---|---|---|---|---|---|---|---|---|---|
| | +property | | −property | | +property | | −property | | +property | | −property | |
| | $\Delta \uparrow$ | $\% \uparrow$ | $\Delta \downarrow$ | $\% \uparrow$ | $\Delta \uparrow$ | $\% \uparrow$ | $\Delta \downarrow$ | $\% \uparrow$ | $\Delta \uparrow$ | $\% \uparrow$ | $\Delta \downarrow$ | $\% \uparrow$ |
| 0 | 1.19 | 89.3 | -0.83 | 74.2 | 1.39 | 95.2 | -0.33 | 65.2 | 0.46 | 72.4 | -1.66 | 87.3 |
| 0.01 | 0.98 | 83.8 | -0.65 | 70.9 | -0.30 | 32.1 | -0.49 | 76.5 | 0.30 | 62.8 | -1.54 | 84.7 |
| 0.05 | 1.35 | 94.0 | -1.12 | 81.4 | 1.39 | 95.2 | -0.42 | 69.6 | 0.52 | 73.4 | -1.27 | 79.7 |
| 0.1 | 1.48 | 94.0 | -1.32 | 85.6 | -0.06 | 48.9 | -0.44 | 72.7 | 0.42 | 67.6 | -1.42 | 82.5 |
| 0.5 | 0.51 | 70.3 | -0.89 | 75.1 | 0.13 | 58.1 | -0.38 | 72.5 | 0.21 | 58.5 | -1.33 | 81.0 |
| 1.0 | 0.73 | 77.6 | -0.66 | 69.3 | 1.33 | 93.5 | -0.87 | 86.9 | 0.44 | 68.9 | -1.61 | 85.5 |

# D    ADDITIONAL OPTIMIZATION TRAJECTORIES

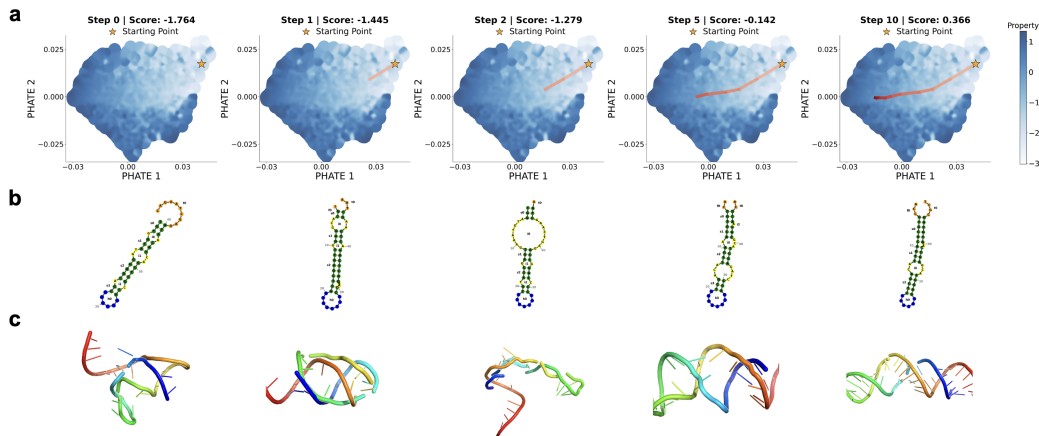

Figure S1: More examples of latent space trajectories. **(a)** Trajectories in the PHATE space. **(b)** 2D structures. **(c)** 3D structures.

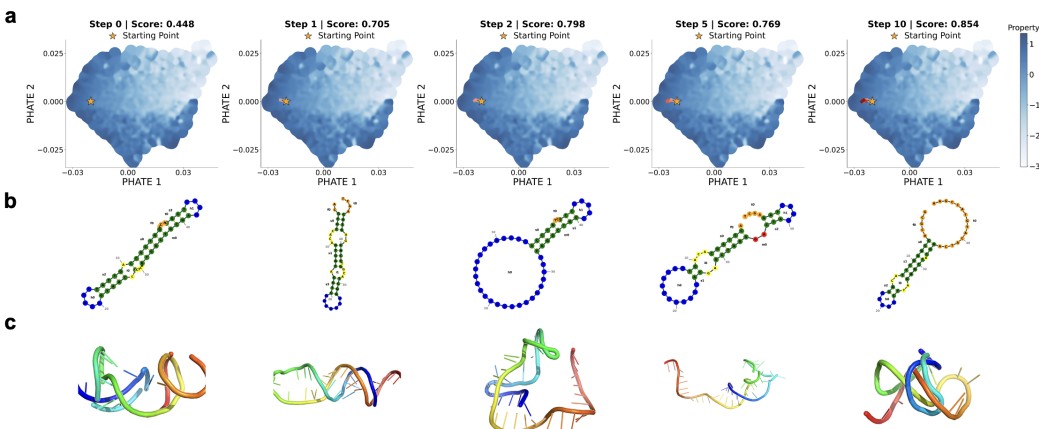

Figure S2: More examples of latent space trajectories. **(a)** Trajectories in the PHATE space. **(b)** 2D structures. **(c)** 3D structures.

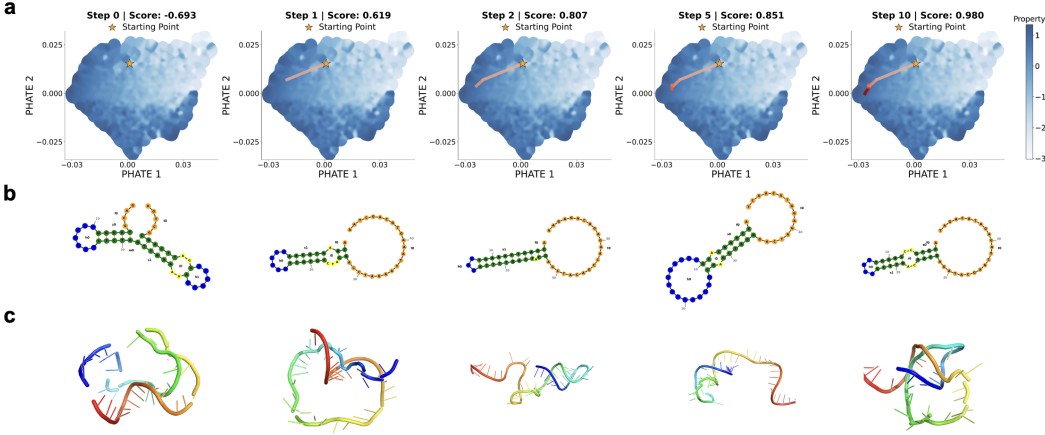

Figure S3: More examples of latent space trajectories. **(a)** Trajectories in the PHATE space. **(b)** 2D structures. **(c)** 3D structures.

# E  EXTENDED RELATED WORKS

**Sequence-to-function modeling**  A central goal in biological sequence modeling is predicting quantitative properties (e.g., expression level, stability) directly from the sequence (Oliver, 1996). Recent deep learning models trained on high-throughput experimental data have demonstrated strong performance in this setting, particularly for regulatory regions such as 5' UTRs and promoters (Sample et al., 2019; Vaishnav et al., 2022). Models such as ConvNets (Chen et al., 2024) and Transformers (He et al., 2023a) have been used to capture complex dependencies in mRNA space, and form the basis for downstream prediction of properties.

**Generative models for design**  Generative models enable sampling of novel sequences enriched for desired traits. Variational autoencoders (VAEs) (Kingma et al., 2013) have been applied to proteins to learn smooth latent spaces that are amenable to gradient-based optimization (Sinai et al., 2017; Castillo-Hair et al., 2024). ProteinMPNN (Dauparas et al., 2022), although described as a message-passing neural network by the authors, shares core design principles with autoencoders. Generative adversarial networks (Goodfellow et al., 2020) such as Méndez-Lucio et al. (Méndez-Lucio et al., 2020) or ProteinGAN (Wu et al., 2021) and autoregressive language models such as ProGen (Madani et al., 2023) have also been used to generate diverse protein sequences. More recently, diffusion models (Ho et al., 2020) have shown promise in discrete domains. For example, RFdiffusion (Watson et al., 2023) generates proteins unconditionally or conditioned on structural constraints. These methods can be readily adapted to mRNA design.

**Optimization of biological sequences**  Sequence optimization can be framed as a black-box search or a differentiable surrogate-guided process. Several approaches relax discrete inputs for gradient-based updates, such as using straight-through estimators (Linder et al., 2019). ReLSO learns a continuous latent space and performs gradient ascent (Castro et al., 2022). Others apply reinforcement learning (Eastman et al., 2018) or Monte Carlo algorithm (Wirecki et al., 2023) for sequence optimization. Methods such as Fast SeqProp (Linder & Seelig, 2021) and LaMBO (Stanton et al., 2022) have demonstrated success in optimizing sequences under multi-objective constraints.

**Integration of structural context**  While the present work strictly focuses on the mRNA sequence, many successful models incorporate inductive biases from the structures. ProteinMPNN (Dauparas et al., 2022) and diffusion-based inverse folding (Yi et al., 2023) condition sequence generation on 3D structures. ImmunoStruct (Givechian et al., 2025) jointly models protein sequence, structure, and biochemical properties to predict immunogenicity. CellSpliceNet (Afrasiyabi et al., 2025) integrates long-range sequence, local regions of interest, secondary structure, and gene expression to predict alternative slicing. EternaFold (Wayment-Steele et al., 2022) incorporate predicted secondary structures to improve fitness prediction. Although in our work we did not incorporate mRNA structures, extending `RNAGenScape` to sequence-structure joint modeling and optimization could be a promising direction.

