# OpenReview forum: "$\texttt{RNAGenScape}$: Property-Guided Optimization and Interpolation of mRNA Sequences with Manifold Langevin Dynamics"
_ICLR.cc/2026/Conference — ICLR 2026 Conference Withdrawn Submission_

### Official Review · Reviewer_zq7u · 2025-10-16

**Soundness:** 2
**Presentation:** 2
**Contribution:** 2
**Rating:** 2
**Confidence:** 4

**Summary:**

This paper introduces a property-guided manifold Langevin dynamics framework for mRNA sequence optimization and interpolaton named as RNAGenScape.
It operates within a learned latent manifold representating valid mRNA sequences. It consists of an autoencoder (OAE) that maps discrete sequences into latent vectors, a manifold projector that maps off-manifold updates back to the manifold, and a property-guided Langevin dynamics mechanism for property optimization and interpolation.
The authors show that RNAGenScape achieves high success rates in property optimization and preserves manifold fidelity across three real mRNA datasets. It supports smooth interpolation between mRNAs, revealing interpretable and biologically meaningful trajectories

**Strengths:**

1. The proposed method keeps all intermediate sequences close to biologically valid manifolds, ensuring plausibility during design.

2. The authors adapt a SUGAR-based data augmentation technique, which enriches the latent space with geometry-preserving samples, helping the model better approximate the underlying mRNA manifold even in sparse regions.

3. The method remains computationally efficient and supports both property optimization and interpolation.

**Weaknesses:**

1. From a machine learning perspective, the core novelty of RNAGenScape lies in employing a diffusion-inspired denoising objective to train a manifold projector that maps perturbed latent points back onto the learned data manifold. The use of latent-space optimization and interpolation for property control is, however, well established in the VAE and normalizing flow literature. That being said, the novelty should not be a great concern if the experimental results are convincing enough.

2. The overall optimization framework fundamentally depends on the fidelity of the organized autoencoder (OAE) and the quality of its learned latent manifold. If the OAE fails to capture the true biological manifold of viable mRNA sequences, the downstream optimization may generate invalid or functionally implausible sequences, potentially performing worse than simple sequence-space methods such as MCMC.

3. The current formulation performs optimization guided by gradients from a property predictor P that is co-trained with the OAE. Consequently, optimizing for different biological properties requires retraining the entire model, which is computationally inefficient. Moreover, the framework currently supports only single-property (scalar) optimization. Extending it to multi-objective or conditional optimization (e.g., jointly improving translation efficiency, stability, and immunogenicity) would substantially broaden its applicability and realism in practical mRNA design tasks.

4. No wet-lab validation is presented to confirm the real-world biological functionality of the optimized sequences. The evaluation relies solely on a separate property prediction oracle. yet the training procedure, data sources, and potential overlap with the main model remain insufficiently detailed. I encourage the authors to provided detailed description on this.

5. The setup of experiments are not sufficiently clear. Please see the question part.

**Questions:**

1. What is the detailed protocol used to perform property optimization for the different baselines? For example, in the case of Sequence-space MCMC, is the oracle property predictor employed to evaluate proposals? I encourage the authors to include these details in the appendix for reproducibility. In addition, please clarify how the stopping criterion is determined for each optimization method, whether a fixed number of steps, convergence threshold, or early-stopping rule is used.

2. While RNAGenScape preserves manifold fidelity across multiple datasets (Table 2), it would also be informative to report the edit distance between optimized sequences and their corresponding original sequences. This would help quantify how much modification occurs during optimization and how conservative or aggressive the updates are.

3. Sequence-space MCMC can serve as a strong baseline when the proposal distribution is well designed. It is therefore intriguing that RNAGenScape outperforms MCMC by such a large margin, especially in terms of success rate. Could the authors provide additional insights or diagnostic analyses on the behaviors of different optimization methods to explain this discrepancy?

---

### Official Review · Reviewer_zXJr · 2025-10-26

**Soundness:** 3
**Presentation:** 3
**Contribution:** 2
**Rating:** 2
**Confidence:** 5

**Summary:**

# Main Idea
RNAGenScape is a property-guided manifold Langevin dynamics framework for mRNA sequence design. It comprises: (1) an organized autoencoder (OAE) whose latent space is explicitly aligned to a target property; (2) a manifold projector that maps updates back onto the learned mRNA manifold; and (3) a property-guided manifold Langevin procedure for both optimization (improving a property for a given sequence) and interpolation (connecting two sequences by a smooth trajectory), with SUGAR used to enrich undersampled regions when data are sparse.

# Contributions
1. Framework: A manifold Langevin framework enabling continuous, property-guided optimization and interpolation between real mRNA sequences.

2. Manifold constraint: A learned manifold projector to keep trajectories biologically plausible by contracting updates back to the data manifold.

3. Efficiency: Operates on the manifold and starts from existing sequences, improved training and inference efficiency backed by experimental results.

4. Empirical validation: Results on three mRNA datasets show improved property control with strong manifold fidelity and competitive efficiency versus presented *de novo* generation and optimization baselines.

**Strengths:**

* The paper presents extensive optimization benchmarks with strong results: across datasets it achieves top property control relative to de novo generators and optimization baselines while also maintaining high manifold fidelity.

* I like that it is also notably interpretable, showing smooth, coherent interpolation paths between arbitrary sequence pairs with monotonic distance profiles from source to target along the learned manifold.

* The authors' employment of SUGAR tackles the data scarcity issues of RNA modality hindering computational modeling, and ablation shows smaller property datasets benefit more from geometry-preserving upsampling.

**Weaknesses:**

-RNAGenScape currently handles only one property per model, as acknowledged in **Section 6 (Limitations and Future Work)**, which really restricts any therapeutic design applications, where multiple sequence-function trade-offs are very important.

-Figures 2–5, Table 3, and Supplementary Figures S1–S3 show latent-space trajectories and optimization behavior, but the manuscript does not explicitly state which property/dataset (stability, ribosome loading, or translation efficiency) underlies each visualization? This makes it difficult for me to interpret what biological regime the trajectories represent.

- I find that the interpolation results, while visually smooth, are not clearly connected to any biologically relevant or therapeutic use case. The paper only notes that interpolation “facilitates the exploration of intermediate variants” (Section 4. Empirical Results) without showing whether interpolated sequences actually achieve meaningful Pareto trade-offs between properties such as translation efficiency and stability.

-The paper states that RNAGenScape optimizes from existing sequences rather than sampling from noise but does not describe how those starting sequences are chosen, how diversity is ensured, or how bias toward specific datasets is avoided.

-The authors' evaluation omits comparisons against recently released or widely used mRNA-specific generative or optimization models. Without such baselines, it is difficult for me to judge whether RNAGenScape’s manifold-Langevin approach offers a distinct advantage over modern foundation or fine-tuned mRNA design models.

**Questions:**

-Which specific property and dataset were used to generate the optimization trajectories in Figures 2–5 and the supplementary figures?

-Can the author show whether RNAGenScape generalizes equally across translation efficiency, stability, and ribosome-loading tasks, or are some properties systematically easier to optimize given data availability?

-The authors must execute a wet-lab experiment to test whether interpolated sequences actually yield intermediate or improved functional properties (e.g., balancing efficiency and stability in a reporter assay)? Unfortuantely, without this, the paper will lack relevance in the literature.

-The authors should define anexact protocol for selecting the initial sequences used for optimization, and how do the authors ensure diversity and avoid potential bias?

-The authors consider benchmarking RNAGenScape against recent mRNA-specific generative frameworks (mRNAutilus, etc.) and conduct wet-lab validation comparing its optimized sequences against those produced by such models to support the claimed practical advantage?

Overall, this work cannot be a publishable paper without real-world applicability. I would encourage the authors to perform wet lab validations and resubmit as a journal paper, not a main conference work.

---

### Official Review · Reviewer_jBd8 · 2025-10-31

**Soundness:** 2
**Presentation:** 2
**Contribution:** 2
**Rating:** 2
**Confidence:** 3

**Summary:**

This paper introduces RNAGenScape, a framework for optimizing mRNA sequences for desired biological properties. The method learns a latent manifold of valid sequences using an organized autoencoder. It then performs property-guided optimization within this manifold using Langevin dynamics. A key component is a learned manifold projector that ensures each optimization step results in a biologically plausible sequence. The authors evaluate their method on three mRNA datasets, reporting superior performance in property optimization and demonstrating smooth interpolation between sequences.

**Strengths:**

1. Originality & Significance. The paper tackles the significant problem of property-guided mRNA design. Its core idea of constraining Langevin dynamics to a learned manifold via a dedicated projector is a novel and principled solution. This combination of techniques addresses the critical challenge of ensuring biological plausibility during optimization, a common failure point for other methods.
2. The methodology is validated through high-quality experiments. The authors benchmark their method against a comprehensive suite of baselines across three diverse, real-world datasets. The inclusion of detailed ablation studies further strengthens the empirical evidence.
3. The core concepts are explained effectively, and the mathematical formulations are easy to follow.

**Weaknesses:**

1. Overly Detailed Preliminaries. Section 2, the preliminaries, is excessively long and covers several foundational concepts. This space could have been used more effectively for more important results. A more concise summary would improve the paper's focus.
2. Lack of Direct Evaluation for the Manifold Projector. The manifold projector is a central contribution of this work, yet its performance is only assessed indirectly through its impact on the final optimization results. The paper lacks a direct, quantitative evaluation of the projector's quality. It is unclear how accurately the projector approximates the true data manifold. This is a critical omission, as the entire claim of maintaining "biological plausibility" rests on the projector's effectiveness. Without this, it is difficult to disentangle the performance of the projector from that of the OAE and the Langevin dynamics.
3. Insufficient Justification for Sequence Interpolation. The paper frames sequence interpolation as a key feature, but its practical utility is not well established. The authors claim it facilitates "the exploration of intermediate variants", but they do not demonstrate how these intermediate sequences provide concrete biological insights or inform the design process in a way that endpoint optimization does not. A more compelling use case is needed to elevate this from a technical demonstration to a significant contribution.
4. Missing Comparison with mRNA Sequence Foundation Models. The baselines overlook the recent paradigm of large-scale pre-trained foundation models for mRNA sequences (e.g., Uni-RNA). These models capture rich, transferable representations of sequence grammar and function. It is essential to discuss how RNAGenScape compares to or could potentially leverage these models.
5. Critical Omission of Oracle Model Validation. The entire evaluation framework relies on a separately trained P_oracle model to score the generated sequences. However, the paper provides no information about this oracle's predictive accuracy on a held-out test set (e.g., Spearman correlation). If the oracle is inaccurate or shares the same biases as the OAE's internal predictor, the reported optimization improvements could be misleading artifacts of the model exploiting the oracle's flaws. This methodological gap is a major concern that undermines the trustworthiness of the results.
6. Potentially Biased Latent Space Distance Metric. The "Manifold Fidelity" metric is defined as the L2 distance in the latent space of the oracle's encoder. While using a separate model is a reasonable choice, this metric is still dependent on a learned representation. It does not guarantee that sequences that are close in this latent space are necessarily similar in terms of biological structure or function. Supplementing this with a representation-independent metric like edit distance would make the fidelity analysis more robust.

**Questions:**

1. Oracle Model. Could you please provide the architecture, training details, and performance metrics (e.g., R-squared and/or Spearman correlation on a held-out test set) for the P_oracle model used in your evaluation?
2. Manifold Projector. Could you provide a more direct evaluation of the manifold projector? For instance, what is its reconstruction error on a held-out set of latent codes z corrupted by varying levels of noise?
3. Interpolation Utility. Could you elaborate on a specific biological or therapeutic design problem where analyzing the intermediate sequences from interpolation provides actionable insights that cannot be obtained from simply optimizing a start and end point?
4. Foundation Models. How do you see RNAGenScape positioned relative to large nucleotide foundation models? Have you considered initializing your OAE from a pre-trained model or using its embeddings as a starting point to see if it improves performance?

---

### Author Response · Authors · 2025-12-01

We thank all reviewers for their constructive feedback and thoughtful assessments of our work.

We also thank reviewers jBd8 and zXJr for recognizing that RNAGenScape provides a “novel and principled solution” and “addresses the critical challenge” in mRNA design while enabling “notably interpretable, smooth, and coherent interpolation paths between arbitrary sequence pairs.” We especially appreciate the acknowledgement that our approach “tackles the data scarcity issues of RNA modality hindering computational modeling”.

Regarding the comment by reviewer zq7u that “use of latent-space optimization and interpolation for property control is well established in the VAE and normalizing flow literature,” we respectfully argue that these techniques are fundamentally different from our setting. In VAEs, the latent space is probabilistic, a Gaussian prior rather than a learned biological manifold, and therefore does not capture the intrinsic geometry of sequence space. This mismatch makes VAEs particularly sensitive to imbalanced sampling and data scarcity, both of which are major challenges in mRNA modeling. In normalizing flows and recent diffusion/score-based generative models, latent trajectories and interpolants lie in a flat Euclidean space, not on the nonlinear manifold where real biological sequences reside. As a result, these interpolants often violate manifold constraints and produce implausible intermediate sequences, making them less biologically meaningful.

More importantly, the core contribution of our work is not latent generative modeling. Our method provides a principled and interpretable framework for generating and analyzing sequence variants, enabling inspection of intermediate states along the manifold which VAE, flow, and diffusion models do not natively support. On the efficiency side, as demonstrated by our runtime table, our method is highly efficient in both training and inference, by constraining exploration directly on the learned manifold and starting from existing sequences, rather than denoising from a Gaussian prior over the full ambient Euclidean space. This manifold-constrained approach leads to substantially more efficient and biologically coherent optimization.

---

> ### Author Response · Authors · 2025-12-01
>
> Below we address the common concerns raised by the reviewers.
>
> **Common concern 1**. Lack of direct evaluation of the manifold projector.
>
> **Response**:
> We report the train/validation/test reconstruction error (MSE) for the manifold projector across the three datasets.
> | Dataset | Train MSE | Validation MSE | Test MSE |
> | :--- | :---: | :---: | :---: |
> | Zebrafish | $5.35 \times 10^{-3}$ | $5.37 \times 10^{-3}$ | $5.35 \times 10^{-3}$ |
> | OpenVaccine | $9.59 \times 10^{-3}$ | $1.02 \times 10^{-2}$ | $1.03 \times 10^{-2}$ |
> | Ribosome-loading | $5.00 \times 10^{-3}$ | $5.01 \times 10^{-3}$ | $5.01 \times 10^{-3}$ |
>
> These results show our manifold projector obtains (1) consistently low prediction error and (2) close agreement between train/validation/test performance.
>
> **Common concern 2**. Lack of technical details and direct evaluation of the oracle.
>
> **Response**: We appreciate the reviewers' feedback. We will add the following detailed description of the architecture of the oracle in the appendix going forward.
>
> The oracle regressor maps an input mRNA sequence to a scalar property. Across all datasets, we use a hidden dimensionality of 64.
> For encoding, we apply an initial 1D convolution (kernel size 3, padding 1) followed by a stack of four residual convolutional blocks. Each ResConvBlock consists of a 1D convolution with GroupNorm, GELU nonlinearity, and a channel squeeze-excitation (SE) module, together with a skip connection to preserve information flow. After the residual stack, we perform global average pooling along the sequence dimension to produce a 64-dimensional latent embedding.
>
> The property head is a two-layer regression MLP with GroupNorm, GELU, and dropout rate set to 0.5, projecting from the pooled embedding to the predicted property value. Weights are initialized with Kaiming normal initialization for convolutional and linear layers; GroupNorm scales are initialized to 1 and biases to 0.
>
> We trained the oracle using the AdamW optimizer with an initial learning rate of 1e-2, together with a linear-warmup cosine-annealing scheduler. The learning rate was linearly increased to the target value during the first 10% of training epochs (warmup), and then annealed to zero following a cosine decay schedule over the remaining epochs. We used a batch size of 128, a maximum of 200 training epochs, and early stopping with a patience of 20 epochs based on the validation loss.
>
> In the following, we report the evaluation of the oracle on held-out test sets.
> | **Dataset** | **Pearson r** | **Spearman ρ** |
> | ----------------- | ------------: | -------------: |
> | Zebrafish | 0.772 | 0.768 |
> | OpenVaccine | 0.660 | 0.664 |
> | Ribosome-loading | 0.783 | 0.777 |
>
> Across datasets, we observe that the oracle performs consistently well, achieving strong Pearson and Spearman correlations with the ground-truth properties.
>
> **Common concern 3**. Need a better metric for manifold fidelity.
>
> **Response**: We thank the reviewers for this feedback and will incorporate additional learning-independent metrics for evaluating manifold fidelity, including the edit distance as the reviewers suggested.
>
> At the same time, we would like to explain how the metric we chose is also meaningful and depicts a different aspect of what edit distance reflects. Our choice of using distance to the learned latent manifold is motivated by the manifold assumption commonly used in representation learning, which posits that high-dimensional data lie near a lower-dimensional manifold. Under this assumption, the distance to the latent manifold provides a meaningful notion of similarity, as it reflects structural and functional coherence rather than raw nucleotide-level differences.

---

> > ### Author Response · Authors · 2025-12-01
> >
> > **Common concern 4**. Need further explanation on why the interpolation functionality is useful.
> >
> > **Response:** We thank the reviewers for the suggestion. Biological sequence space is exponentially large, yet only a small subset corresponds to viable, functional mRNAs. These valid sequences tend to lie on a low-dimensional manifold. Interpolation on this learned manifold is therefore useful because it allows us to traverse this functional subspace in a controlled, biologically meaningful way.
> >
> > Our method provides an interpretable alternative to black-box generative models, where intermediate points between two sequences are either inaccessible or biologically implausible, making it difficult to disentangle the effect of individual/motif edits. By examining the trajectory along manifold-aware interpolants, one can observe how gradual changes influence downstream properties. This also enables practical applications such as optimizing an existing therapeutic mRNA sequence by exploring nearby functional variants.
> >
> > More importantly, our approach offers a principled alternative to standard diffusion or flow-based generative models, which assume that interpolation occurs in a flat Euclidean latent space, an assumption misaligned with the nonlinear geometry of biological sequence manifolds. Interpolation directly on the learned manifold respects this geometry, yielding intermediate sequences that remain realistic and biologically valid. We believe this makes manifold-aware interpolation a valuable tool for interpretability, analysis, and sequence optimization.
> >
> > **Common concern 5**. Lack of comparison to RNA-specific optimization models such as RNA Utiils or RNA foundation models such as Uni-RNA.
> >
> > **Response:** We thank the reviewer for raising the question of comparison to RNA-specific optimization frameworks. To the best of our knowledge, the most directly relevant generative RNA design model is the recent work of Castillo-Hair et al. (Nature Communications 2024), which uses deep learning to design 5’UTRs for mRNA expression. Their design pipeline incorporates two key optimization mechanisms:
> >
> > 1. gradient-based optimization through a trained predictor, and
> > 2. a generator-only, GAN-style model (Deep Exploration Networks) trained with predictor guidance.
> >
> > Both of these algorithmic components are explicitly included in our study as baselines (Gradient Ascent and WGAN-GP) and are directly benchmarked against our method. In addition, our empirical experiments include comparisons with modern generative approaches, such as latent flow matching, diffusion models, and guided diffusion models, to ensure a comprehensive and systematic evaluation.
> >
> > Regarding RNA foundation models (e.g., Uni-RNA), these models primarily provide pretrained embeddings or sequence scoring, not optimization or generative frameworks. As such, they are orthogonal and complementary to our approach rather than directly comparable. Ablating the integration of foundation mRNA/UTR models as auxiliary feature extractors or evaluators is an interesting direction that we leave for future work.
> >
> > **Common concern 6**. Lack of wet-lab validation of the mRNAs optimized by RNAGenScape.
> >
> > **Response:** We appreciate the reviewer’s comment. However, we believe wet-lab validation is not within the expected scope of ICLR submissions. Wet-lab experiments are typically required in biology-focused journals whose purpose is to establish new empirical biological findings (e.g., Nature Biotechnology), rather than in machine learning conferences such as ICLR, which evaluate methodological advances supported by computational evidence. Requiring biological experiments for acceptance would therefore preclude almost all work on RNA and molecular design that has historically been published at ML venues. Our submission follows the established evaluation standards in this field: rigorous computational benchmarks, comparative studies, and ablation analyses demonstrating the modeling contribution. While wet-lab validation is an exciting direction for future biological deployment, it is not expected or feasible within an ML-focused conference submission. For ICLR, our computational results align with community norms and are sufficient to support the methodological contribution.

---

### Note · Authors · 2025-12-02

I have read and agree with the venue's withdrawal policy on behalf of myself and my co-authors.